# Contextual Inferences, Nonlocality, and the Incompleteness of Quantum Mechanics

**DOI:** 10.3390/e23121660

**Published:** 2021-12-10

**Authors:** Philippe Grangier

**Affiliations:** Laboratoire Charles Fabry, Institut d’Optique Graduate School, Centre National de la Recherche Scientifique (CNRS), Université Paris Saclay, 2 Avenue Augustin Fresnel, F91127 Palaiseau, France; philippe.grangier@institutoptique.fr

**Keywords:** quantum mechanics, contextuality, inferences, nonlocality

## Abstract

It is known that “quantum non locality”, leading to the violation of Bell’s inequality and more generally of classical local realism, can be attributed to the conjunction of two properties, which we call here elementary locality and predictive completeness. Taking this point of view, we show again that quantum mechanics violates predictive completeness, allowing the making of contextual inferences, which can, in turn, explain why quantum non locality does not contradict relativistic causality. An important question remains: if the usual quantum state ψ is predictively incomplete, how do we complete it? We give here a set of new arguments to show that ψ should be completed indeed, not by looking for any “hidden variables”, but rather by specifying the measurement context, which is required to define actual probabilities over a set of mutually exclusive physical events.

## 1. Introduction

After many years of theoretical and experimental research, it can be now said that the door has been closed on the historical Einstein and Bohr’s quantum debate [1,2,3]. On the way, this research opened the door to many new ideas and experiments, leading ultimately to the development of quantum technologies. As a reflection on these evolutions, our point view here is to go back to the Einstein–Bohr debate and to propose answers to the initial questions: Is the “wave function” a complete description of physical reality? What is the role of locality? What about relativistic causality?

We will see that, contrary to what is often said, Einstein, Podolsky, and Rosen were perhaps not so wrong, and Bohr not so right—and that some lessons may be learned regarding what quantum mechanics is telling us about physical reality.

Our reasoning uses the idea of contextuality, which is currently an extremely active field of research, connected with many foundational issues [4,5,6,7,8,9,10,11]. However, rather than pursuing these interesting lines of research, we step back to discussions from the 1980s [12,13,14], which were perhaps too quickly dismissed. This is because fully exploiting them amounts to admitting that the usual |ψ〉 is incomplete, which is a shocking statement that was rejected by Bohr himself in 1935 [3].

However, many things have happened since then, especially with regards to contextuality and nonlocality. Thus, we propose a “not so shocking” way to complete |ψ〉: very schematically, it tells that a usual state vector is incomplete as long as a complete set of commuting operators admitting this vector as an eigenstate has not been specified. More details are given below as well as how to use this idea for our purpose.

## 2. Probabilistic Framework

We use a general framework for conditional probabilities as presented, for instance, by E.T. Jaynes in [12], and also related to the analysis in [13,14]. We emphasize that these calculations are quite general and do not imply any commitment to a specific view on probabilities—Bayesian or otherwise. The equations we write apply both to usual quantum mechanics and to local hidden variable theories (LHVT), and the main interest of this calculation is to show explicitly where these two descriptions split, and why. We will also indicate when some (reasonable but not mandatory) hypotheses will be made.

We consider the well-known EPR–Bohm–Bell scheme [1,15], where polarization measurements are carried out on entangled photon pairs, described by some quantity λ in a variable space Λ. Alice and Bob carry out measurements defined by the respective polarizers’ orientations *x* and *y* and obtain binary results a=±1 and b=±1.

According to the usual rules of probabilities, and with some care but without loss of generality [12], one can write the following relation between conditional probabilities, by conditioning on λ in some a priori unknown hidden variable space Λ
(1)$$P\left(ab|xy\right)=\sum_{\lambda \in \Lambda }P\left(ab|xy\lambda \right)P\left(\lambda |xy\right)$$

In addition to this purely probabilistic relation, we introduce some requirements about the physics we want to describe, and we do it in the most general way: we assume that usual Quantum Mechanics (QM) and special relativity in the form of Relativistic Causality (RC) are true. We note that being true does not necessarily mean being complete [15,16,17], and we will come back to that issue later on. It should also be clear that theories where *a* and *b* are deterministic functions of λ, *x*, *y* do fit in this probabilistic framework as special cases; however, determinism has important consequences to be discussed below.

## 3. Enforcing Relativistic Causality

A first consequence of RC, sometimes called “freedom of choice”, consists in requiring that λ does not depend on the variables (*x*, *y*) representing Alice and Bob’s choices of measurement settings. In other words, the choices of measurements (*x*, *y*) should not act on the way photons are emitted (λ), since these events are space-like separated. This boils down to the independence condition P(λ|xy)=P(λ), or equivalently P(xy|λ)=P(xy), which is fulfilled by all the theories we are interested in. We note that one may reject freedom of choice by arguing that the random events λ, *x* and *y* are correlated from their distant past, see e.g., [18]. However such “superdeterministic” theories are highly speculative and here we do choose to keep free will. We have thus
(2)$$P\left(ab|xy\right)=\sum_{\lambda \in \Lambda }P\left(ab|xy\lambda \right)P\left(\lambda \right)$$

For a given initial state λ of the pair, a relevant theory should provide P(ab|xyλ), and thus we now focus on this conditional probability. For the sake of clarity, λ is a generic notation to specify whatever may be specified about the emission of the photon pair, in a given shot. This may include variables that fluctuate from shot to shot and other variables that do not. On the other hand, *x*, *y* and λ are not causally related as written above.

We note that Equation (Equation 2) is true also for QM, where the variable space Λ contains only one λ corresponding to the initial state of the entangled pair (e.g., a singlet state). It is standard in recent demonstrations of Bell’s inequalities [19,20] to assume that P(ab|xyλ) is a probability for a given λ, and thus there is no restriction of generality here.

Now, without any further assumptions, one can write from basic rules of inference
(3)$$\begin{array}{ccc} P(ab|xy\lambda ) & = & P(a|xy\lambda )P(b|xy\lambda a)\\ & = & P(a|xy\lambda b)P(b|xy\lambda ) \end{array}$$
where the two decompositions refer, respectively, to Alice and Bob, and where on Alice’s side
P(a|xyλ)= probability of Alice obtaining result *a* for input *x*, andP(b|xyλa)= probability of Bob obtaining result *b* for input *y*, calculated by Alice who knows *x* and *a*;
whereas, on Bob’s side,
P(b|xyλ)= probability of Bob obtaining result *b* for input *y*, andP(a|xyλb)= probability of Alice obtaining result *a* for input *x*, calculated by Bob who knows *y* and *b*.

Clearly, a meaningful requirement in Equation (Equation 3), again related to RC, is that the choice of measurement by Alice (resp. Bob) should not have an influence on the result by Bob (resp. Alice). This implies that P(a|xyλ)=P(a|xλ) and P(b|xyλ)=P(b|yλ), and we call this condition “elementary locality” (EL), meaning that it is fulfilled for each given λ. As a consequence, one has
(4)$$\begin{array}{ccc} P(ab|xy\lambda ) & = & P(a|x\lambda )P(b|y\lambda ,xa)\\ & = & P(a|x\lambda ,yb)P(b|y\lambda ) \end{array}$$
where, in general, one cannot remove xa from P(b|yλ,xa), nor yb from P(a|xλ,yb).

Let us emphasize that, so far, we have respected QM and RC at each step, and it can easily be checked that Bell’s inequalities cannot be obtained from Equation (Equation 4). Correspondingly, if interpreted “à la Bell”, keeping xa and yb in Equation (Equation 4) looks like an influence of one measurement on the other side. Yet, this conclusion is not warranted since Alice calculates a probability for Bob’s result by using only what is locally available to her (resp. him by switching Alice and Bob); this does not influence in any way what is happening on the other side.

## 4. Contextual Inferences vs. Bell’s Hypotheses

We conclude that, given Equation (Equation 4), there is still a missing step to reach Bell’s theorem. In order to identify it, let us recall that locality “à la Bell” can be seen as a conjunction of two conditions:

- The first condition is “elementary locality” (EL), already spelled out above:

(EL) P(a|xyλ)=P(a|xλ) and P(b|xyλ)=P(b|yλ), taken as true as explained before.

- A second condition—let us call it “predictive completeness” [14]—is given by:

(PC) P(a|bxyλ)=P(a|xyλ) and P(b|axyλ)=P(b|xyλ), and this is interpreted physically below. Taken together, the conditions (EL) and (PC) entail the factorization condition P(ab|xyλ)=P(a|xλ)P(b|yλ) and, therefore, lead to Bell’s inequalities [19,20].

These two conditions have been given various names, e.g., parameter independence for (EL) and outcome independence for (PC), see [21]; here, we stay closer to Jarrett’s definitions [13,14], though some of our conclusions are different. We note also that (EL) is not accepted by Bohmian mechanics (BM), which agrees with relativistic causality only after averaging on λ. Thus our initial requirement of fulfilling RC at every step is not compatible with BM; see also the Conclusion section.

In order to justify the hypothesis (PC) and the wording “predictive completeness”, one must emphasize that Bell’s factorization condition P(ab|xyλ)=P(a|xλ)P(b|yλ) relies on the idea that λ specifies everything that can be known about the pair of particles; given this assumption, condition (PC) should be obvious, because knowing xa cannot bring anything more to Alice’s probability calculation; hence, the name of predictive completeness. For instance, theories where *a* and *b* are deterministic functions of λ, *x*, *y* must satisfy (PC).

On the other hand, (λxa) occurring in the probability P(b|yλ,xa)=P(b|y,λxa) is not a property carried by Bob’s particle, but it involves both the properties of Bob’s particle (included in λ) and the result of Alice’s measurement (described by xa). In other words, (λxa) refers to a property of Bob’s particle, not in and by itself, but *within a context* defined by Alice’s result. In the CSM language [22,23,24,25,26,27,28], (λxa) defines a modality for Bob’s particle, in Alice’s context. The correspondance between the respective modalities (λxa) and (λyb) can only be probabilistic, with probability 1 if a=b [24]. Alice’s context and result cannot have any influence on Bob’s particle, and they do not, since (λxa) is only used locally by Alice according, again, to Equation (Equation 4).

Given this situation, we take a major new step beyond the previous discussions by admitting that the description given by λ (or ψ in the quantum case) is incomplete indeed, and that knowing (xa) does bring something new to Alice. Then, condition (PC) can be violated, by **Alice making a “contextual inference” about Bob’s result.** In order to make sense of this idea, it is essential to realize that (i) contextual inference is a non-classical phenomenon, and (ii) it agrees with relativistic causality, as we explain now.

(i) In classical physics, condition (PC) as defined above is verified, and Bell’s factorization condition follows. However, in quantum physics, knowing Alice’s measurement and result allows her to predict more, without invoking any action at a distance. This is because λ≡ψ does not tell which measurements will be actually carried out by Alice and Bob—in other words, λ≡ψ is predictively incomplete [13,17].

Adding this information where and when it is **locally** available improves Alice’s prediction about Bob’s result, and Bob’s about Alice’s, in agreement with Equation (Equation 4), showing the suitability of the concept of contextual inference. This effect does not show up in classical physics, because a classical λ is complete; however, it does show up in QM because a quantum ψ is (predictively) incomplete, as long as a measurement context has not been specified (for more details, see [17] and the last sections below).

(ii) Since contextual inference only applies to probabilities appearing in Equation (Equation 4), it does not involve any physical interaction outside light cones; therefore it obeys relativistic causality. A typical wrong line of thinking would be to say: if Alice can predict with certainty some results by Bob (perfect correlations, obtained when a=b), then either Bob’s result is predetermined, or there are instantaneous actions at a distance. However, this dilemma only applies in a classical framework, where particles’ properties are defined in an absolute way, and Bell’s inequalities do apply. In a quantum framework, Alice can make local inferences by using additional information that is available to her, e.g., (λxa) in the above example; and these predictions can only be checked by accessing Bob’s results afterwards, in a local and ordinary causal way.

## 5. Discussion

It is also interesting to draw a standard light-cone picture (see Figure 1), in order to show explicitly how contextual inference may be used when the relevant information is locally available. More precisely, this diagram allows us to separate on the one hand the localized events in space time (first the production of λ, *x* and *y*, and then the separated read-out of *a* by Alice, and *b* by Bob), and on the other hand the corresponding predictions, that are inferences, not influences, and thus no “action at a distance” is involved.

Another remark may be useful: as suggested by the light cones pictures in Figure 1, one may consider that *x* and *y* are also issued from independent random processes in variable spaces *X* and *Y*, as it is done in loophole-free Bell tests [1]. Then, the global probability writes
(5)$$P\left(xa,yb\right)=\sum_{\lambda \in \Lambda ,x\in X,y\in Y}P\left(ab|xy\lambda \right)P\left(\lambda \right)P\left(x\right)P\left(y\right)$$
where P(ab|xyλ) is given by Equation (Equation 4) as before. Taking Λ={λ}, $X=\{x_{1},x_{2}\}$ with $P\left(x_{1}\right)=P\left(x_{2}\right)=1/2$, $Y=\{y_{1},y_{2}\}$ with $P\left(y_{1}\right)=P\left(y_{2}\right)=1/2$ as in a usual Bell test, one finds P(xa,yb)=P(ab|xyλ)/4.

Correspondingly, the random variable (xa,yb) may take 16 mutually exclusive values, not 4, and Bell’s inequalities cannot be written anymore. Bell’s reasoning requires calculating the correlation functions $E\left(x,y\right)={\langle ab\rangle }_{x,y}$ by using P(ab|xy), not P(xa,yb), and thus the four different measurements apply to the same sample space Λ. This means implicitly that λ completely carries the pairs’ properties (and the measurement results can be predicted from the knowledge of λ alone), as it would be the case in classical physics.

However, this is counterfactual [15,16] with respect to the quantum approach, where $\left\{\lambda \equiv \psi ,x_{1},y_{1}\right\}$, $\left\{\lambda \equiv \psi ,x_{1},y_{2}\right\}$, $\left\{\lambda \equiv \psi ,x_{2},y_{1}\right\}$, $\left\{\lambda \equiv \psi ,x_{2},y_{2}\right\}$ are four different situations that should not be merged within a single *S* value, contrary to Bell’s reasoning [19,20]. This is another way to tell that ψ is not complete, and requires a context specification to be turned into an actual probability distribution.

*Completing ψ?—* A second major new step is to answer the question: If ψ is not complete, does it tell anything concrete by itself? It does, because it indicates a set of contexts, corresponding to all the observables, including ψ as an eigenvector, where the associated measurement result (eigenvalue) is predictable with certainty. In recent papers [23,24,25,26,27], we introduced a framework that makes a careful distinction between the usual ψ without a context and the physical state within a context—called a modality (see also Appendix A).

In this language, ψ is associated with an equivalence class of modalities, called an extravalence class [25]; whereas the modalities are complete because they are properties of a system within a context; ψ is not, because the context is missing by construction. This gives a nice outcome to the Einstein–Bohr debate, by confirming the incompleteness of ψ [2], and by telling also how to complete it: one should add the context—this actually fits with the “very conditions” required by Bohr’s answer [3,15,28].

## 6. Conclusions

In summary, the violation of Bell’s inequalities by quantum theory and experiments [1] can be explained if one takes into consideration contextual inferences, and these, in turn, are ultimately allowed by the predictive incompleteness of the quantum state: finding actual probabilities for measurement results requires specifying a measurement context. In practice, contextual inferences correspond to what is usually called “quantum non locality”; however, they are not related to locality in a relativistic sense, but rather to the specifically quantum condition that requires attributing physical properties to systems within contexts. The implications on the (in)completeness of QM are discussed in more detail in [17]; however, a few comments are in order:In the above, we argue that ψ is predictively incomplete, but not that QM is incomplete in the sense of being erroneous. There are many practical ways to complete it, by reintroducing the context either “by hand” (like in usual textbook QM) or in a more formal way using algebraic methods [17].The predictive incompleteness of ψ is general, and not limited to entangled states. This is because the measurement context is required to find actual probabilities, or said otherwise, one cannot define a full consistent set of **classical** probabilities applicable to any result in any context. In the language of [29], ψ provides mathematical q-probabilities without interpretation, whereas ψ completed by the specification of the measurement context provides true probabilities for mutually exclusive events.In this article, we enforced (EL) at the beginning and explained how (PC) can be violated by a non-deterministic theory, without any conflict with RC. On the other hand, deterministic theories must agree with (PC), and therefore have to violate (EL) to be compatible with the observed violation of Bell’s inequalities; an example of such a theory is Bohmian mechanics. Generally speaking, if (EL) is rejected, more care must be taken in order to avoid an explicit violation of special relativity [30].Here, we considered the standard version of Bell’s theorem, but many other inequalities may be obtained in the general framework of “local realism”. It would be interesting to look whether the violation of such inequalities is generally due to a violation of (PC); this may be the topic for further work (see Appendix B for three-particle entanglement).

Finally, it is interesting to note that, in [12], Jaynes did not spell out either the “nonlocal” or the “incompleteness” option, though he did all the calculations above. In our opinion, this is because he could not give up the classical idea that particles should be described independently of their contexts. In order to admit the idea of contextual inference, an intellectual quantum jump is required to accept that, in quantum mechanics, one has to take into consideration both the systems **and** the contexts in which they evolve. A simple way not to forget this requirement is to postulate that the “object” carrying well-defined properties is a composite: a (quantum) system within a (classical) context [17,22,23,24,25,26,27,28,31].

## Figures and Tables

**Figure 1 entropy-23-01660-f001:**
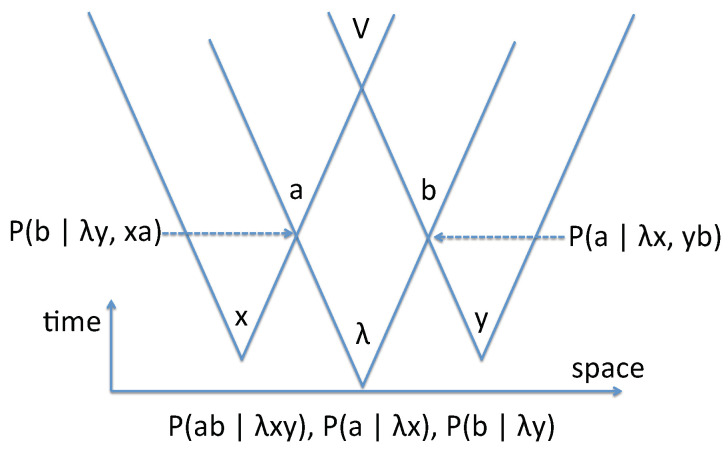
Light-cone picture of the EPR–Bohm–Bell scheme. The photon pair is generated at the bottom of the middle cone, and is described by λ. The measurement settings *x* and *y* are chosen by Alice and Bob in separated light cones. The earliest time for generating the results a|x and b|y are at the intersections of the light cones, and this is also when Alice’s probability P(b|yλ,xa) about Bob’s result, and Bob’s probability P(a|xλ,yb) about Alice’s result become available (dashed arrows). These probabilities result from a contextual inference, which respects relativistic causality and does not entail any action or influence between Alice and Bob. The resulting predictions can be effectively checked in the verification zone V in the common future of all light cones.

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
