# Peer review of "Contextual Inferences, Nonlocality, and the Incompleteness of Quantum Mechanics"

_entropy, 2021, doi:10.3390/e23121660_

Round 1

Reviewer 1 Report

The work provides a sound and interesting view on how  “quantum non locality” does not contradict relativistic causality. Using a general  probabilistic toolkit the authors argue as they themselves put that a state vector incomplete as long as the complete set of commuting operators admitting this vector is provided.

From the viewpoint of scientific relevance, this work is as important and interesting as the EPR paper itself. It is elucidating, pedagogical and technically flawless.

I recommend it for publication in any serious journal of fundamental physics and foundations of quantum theory.

Author Response

We thank the referee for his / her positive comments, and for understanding the main message about the incompleteness of the state vector, and how to complete it. 

Reviewer 2 Report

The paper appears correct and reasonably clear to me in its presentation, but I have some doubts about the originality of its conclusions. The paper supports the claim that the violation of predictive completeness by indeterministic theories (such as standard QM) does not generate a true tension with relativity, whereas a violation of elementary locality by deterministic theories (such as Bohmian mechanics) may display a more serious tension with relativity. As a matter of fact, this claim is totally equivalent to the claims supported by J. P. Jarrett in the paper “On the physical significance of the locality condition in the Bell arguments”,
348 Noûs 18, 569 (1984) [correctly quoted by the author] and A. Shimony in the paper "Controllable and uncontrollable non-locality", in S. Kamefuchi et al. (eds.), Foundations of Quantum Mechanics in the Light of New Technology", Tokyo 1984. Whether the decomposition of the Bell locality condition (factorizability) in terms of two different conditions named in different ways [the authors denotes them as EL and PC], on which the above claim is based, is really useful and plausible by a conceptual and foundational point of view is a different question: the point is controversial and, for instance, T. Maudlin in his book Quantum Non-Locality and Relativity, Wiley 2011, gives some serious arguments to support that the decomposition is not as useful as many have thought. Be the last question as it may, the point is that the main claim of the present paper is quite old, and the use of the new framework of 'contextual inferences' does not change significantly this circumstance. Perhaps an addition might be welcome, in which the author clarifies in a clearer way in which sense his contribution substantially differs from the existing literature.

Author Response

We thank the referee for his / her positive comments.  The article is indeed built upon the argument initially presented by Jarrett, and again by Shimony  with a different terminology, but essentially the same ideas. We take this argument as accepted by the scientific community, and we do not wish to enter into a controversy with Maudlin. The main question by the referee is what we are adding to this old argument.  The answer is actually simple : the conclusion of the argument (with Jarrett’s terminology) is that the state vector psi  is predictively incomplete, but what does that mean in practice ? More precisely, if psi is incomplete, is there a way to complete it, different from the standard hidden variable or supplementary parameters  idea ? And is it possible to show that this predictive incompleteness does not contradict relativistic causality ? 

We bring a fresh answer to these old questions, which is the following :  a state vector is incomplete as long as a complete set of commuting operators (CSCO) admitting this vector as an eigenstate has not been specified.  In other terms, psi must be completed 'from above', by specifying the measurement context or CSCO, and not 'from below', by looking for some hidden variables.  It may be argued that this answer is not new, since it has a clear relationship with Bohr’s answer to the EPR argument; but it also deviates from Bohr, by telling that psi is NOT complete, since the complete specification of the 'state', that we call a modality, requires to specify the context. Our point of view has the merit to acknowledge the contextual aspect of QM, and it can be used to demonstrate in a straightforward way the absence of conflict with relativistic causality, as we do in the paper : the issue at stake is contextual inference indeed, and not nonlocal influence. This point being made, using or not the wording of 'quantum nonlocality' (to speak about contextual inference) is a matter of personal choice that we leave open.

All the above points were already made in the paper, and we do think that they spell out a new point of view about an old debate. But apparently this did not appear clearly to the referee, so we added some sentences at various places in the paper, in order to 'clarify in which sense this contribution substantially differs from the existing literature'. We do hope this will make the article acceptable for publication. 

Round 2

Reviewer 2 Report

I acknowledge the efforts made by the author to emphasize the novelty of the perspective in which the main claims of the paper - in themselves reminiscent of well-knowns claims - are framed. Therefore, I have no objections to publication of the paper in its present form.